# Qualitative and Quantitative Stress Perfusion Cardiac Magnetic Resonance in Clinical Practice: A Comprehensive Review

**DOI:** 10.3390/diagnostics13030524

**Published:** 2023-01-31

**Authors:** Wenli Zhou, Jason Sin, Andrew T. Yan, Haonan Wang, Jing Lu, Yuehua Li, Paul Kim, Amit R. Patel, Ming-Yen Ng

**Affiliations:** 1Department of Radiology, Shanghai Jiao Tong University Affiliated Sixth People’s Hospital, No. 600, Yishan Road, Shanghai 200233, China; 2Department of Diagnostic Radiology, The University of Hong Kong, Hong Kong SAR, China; 3St. Michael’s Hospital, University of Toronto, Toronto, ON M5B 1W8, Canada; 4General Electric, Boston, MA 02210, USA; 5Department of Medicine, University of California San Diego, San Diego, CA 92093, USA; 6Department of Cardiovascular Medicine, University of Virginia, Charlottesville, VA 22903, USA; 7Department of Medical Imaging, HKU-Shenzhen Hospital, Shenzhen 518009, China; 8Department of Diagnostic Radiology, School of Clinical Medicine, The University of Hong Kong, Hong Kong SAR, China

**Keywords:** stress imaging, cardiac magnetic resonance imaging, myocardial ischemia, coronary artery disease, coronary microvascular dysfunction

## Abstract

Stress cardiovascular magnetic resonance (CMR) imaging is a well-validated non-invasive stress test to diagnose significant coronary artery disease (CAD), with higher diagnostic accuracy than other common functional imaging modalities. One-stop assessment of myocardial ischemia, cardiac function, and myocardial viability qualitatively and quantitatively has been proven to be a cost-effective method in clinical practice for CAD evaluation. Beyond diagnosis, stress CMR also provides prognostic information and guides coronary revascularisation. In addition to CAD, there is a large body of literature demonstrating CMR’s diagnostic performance and prognostic value in other common cardiovascular diseases (CVDs), especially coronary microvascular dysfunction (CMD). This review focuses on the clinical applications of stress CMR, including stress CMR scanning methods, practical interpretation of stress CMR images, and clinical utility of stress CMR in a setting of CVDs with possible myocardial ischemia.

## 1. Introduction

Stress CMR is a non-invasive one-stop assessment of myocardial ischemia, myocardial viability, and cardiac function, and it has been widely used for the evaluation of patients with known or suspected CAD due to its excellent diagnostic accuracy [1]. Recently, the 2021 American Heart Association (AHA)/American College of Cardiology (ACC) chest pain guidelines have elevated the role of stress CMR, recommending it as one of the functional testing options for intermediate- to high-risk patients with chest pain with or without known CAD for diagnostic purposes [2]. In addition to diagnosis, stress CMR also provides prognostic value and guides treatment strategies. Myocardial ischemia in patients with chest pain or known significant CAD is associated with the increased risk of major adverse cardiovascular events (MACEs) [3,4,5]. The landmark MR-INFORM trial demonstrated that stress CMR was non-inferior to invasive fractional flow reserve (FFR) in guiding revascularisation strategy in patients with stable angina with regards to MACEs and a reduction in coronary revascularisation [6]. In addition, there is increasing evidence that stress CMR can be utilised for assessing CMD, non-ischemic cardiomyopathy, connective tissue diseases, metabolic syndromes, atrial fibrillation (AF), and other common CVDs [7,8,9,10,11]. In this article, we review the acquisition of stress CMR, the interpretation of myocardial ischemia and viability, and clinical applications in a wide range of CVDs and in children.

## 2. Acquisition of Stress CMR

Dynamic contrast-enhanced perfusion imaging is the basic technique of this exam, which captures the signal changes of contrast passing through the chambers of the heart and myocardium. This method uses electrocardiogram-gated fast T1-sensitive imaging and can be performed both during stress and rest [12]. Typical perfusion sequences used are saturation-recovery with balanced steady-state free precession (SSFP), gradient echo (GRE), or GRE-echo planar hybrid readout. Slice thickness is typically 8–10 mm with temporal resolution around 100–125 ms, usually acquiring every heartbeat, but acquired across two heartbeats in higher heart rates. Myocardial ischemia is assessed during vasodilation, which is commonly induced by injecting adenosine, dipyridamole, regadenoson, or adenosine triphosphate (ATP) [13]. Gadolinium-based contrast is then injected (0.05–0.1 mmol/kg, 3–7 mL/s), followed by a saline flush (≥30 mL) into a peripheral vein after achieving hyperemia to visualise the trajectory of blood flow and myocardial perfusion supplied by normal coronary arteries versus diseased vessels. Ischemic myocardium therefore shows slower perfusion and a decreased T1 signal when compared to normal segments (Figure 1). 

The choice of pharmacologic stress agents is dependent on local preferences. Adenosine increases coronary blood flow approximately three- to five-fold with a short half-life of <10 s, which requires continuous infusion (140 μg/kg body weight/min), and thus needs separate intravenous catheters for the simultaneous injection of adenosine and the contrast agent during the imaging procedure [12,14]. Dipyridamole is given across a 4-min period at a dose of 0.56 mg/kg. Dipyridamole has a longer half-life than adenosine, causing a longer duration of reversible side effects, which is less desirable for patients [15]. Furthermore, dipyridamole has less reproducible vasodilation and inferior results compared to adenosine [7]. Regadenoson has a long half-life of 20 min; it is also widely used due to the convenient non-weight-based fixed dose [13]. The administration of regadenoson removes the need for infusion pumps and two intravenous lines. However, regadenoson also brings the longer persistence of side effects such as dipyridamole, occasionally requiring the use of aminophylline or caffeine to terminate the side effects. ATP has similar hemodynamic mechanisms and a similar vasodilator effect to adenosine, but it usually requires a slightly longer infusion period than adenosine. In addition, it is only used predominantly in the Asian Pacific region because of cost, licensing, and production issues [16,17]. 

Adequate stress should be achieved for an accurate assessment of ischemia. Heart rate (HR) increasing by >10 bpm or systolic blood pressure (SBP) dropping by >10 mmHg, accompanied with clinical symptoms, are commonly used indicators to assess hyperemic response after 2–3 min infusion [13]. A splenic switch-off sign is also a direct marker of adequate adenosine response (Figure 2), which presents as a visual attenuation of splenic perfusion during stress compared with rest. This is because adenosine can act on the A1/A2B receptors on the splenic blood vessels to induce vasoconstriction and reduce the intensity of spleen [18]. In situations in which there is an inadequate stress response, increasing the infusion rates of adenosine up to 210 μg/kg bodyweight/min has been utilised to achieve adequate stress [19]. These higher infusion rates have also been used to overcome the effects of caffeine intake. In terms of rest perfusion imaging, it should be repeated with the same image position and the same dose of gadolinium-based contrast without injecting vasodilator agents after at least 10 min from stress perfusion imaging [13]. During the period, a series of cine CMR images covering the entire ventricles from apex through the base can be acquired to observe the wall motion. Five minutes after the rest perfusion imaging, late gadolinium enhancement (LGE) should be performed to assess myocardial viability. The whole stress CMR exam takes approximately 30 min, and the typical protocol is shown in Figure 3.

## 3. Interpretation of Stress Perfusion CMR

Qualitative interpretation of perfusion imaging is the most convenient method in routine clinical practice [20]. A true perfusion defect is characterised by persistent hypointensity for >5 RR intervals beyond peak myocardial enhancement across more than two pixels. It often appears prominently in the subendocardium and manifests as a transmural gradient across the wall thickness in a coronary distribution [20]. It is also important to consider rest perfusion images to differentiate artefacts from true perfusion defects. The hypointensity presenting on stress images only is more likely a true hypoperfusion due to coronary stenosis. One needs to be aware of dark rim artefacts, which are not true perfusion defects. Dark rim artefacts are typically transient (i.e., <5 R-R intervals), appear in the phase-encoding direction, are one pixel wide, and appear when contrast arrives in the left ventricular (LV) cavity but before myocardial enhancement. 

Myocardial viability assessment requires the LGE images to first classify whether perfusion defects are due to either myocardial ischemia or myocardial infarction. Infarcts seen on LGE typically involve the subendocardium and are consistent with a coronary artery territory (Figure 4). The number of AHA segments involved and the percentage of wall thickness involvement need to be reported to guide clinicians in making revascularisation decisions. Infarcts involving >50% of the wall thickness of an AHA segment are usually regarded as non-viable segments [21]. The transmural extent of LGE predicts the myocardial recovery after successful revascularisation. In Kim et al.’s seminal paper, they showed that in patients with dysfunctional myocardium without any delayed enhancement, there was a high likelihood of recovery (80%). In patients with infarcts on LGE, depending on the wall thickness percentage involved with the infarct, the chance of recovery in cardiac function was 60%, 40%, 10%, and 1% for wall thickness infarcts involving 1–25%, 26–50%, 51–75%, and >75% thickness infarcts, respectively [21]. 

Patients presenting with dysfunctional myocardium but normal LGE images may have underlying myocardial stunning or hibernation with an absence of infarction. In stunned myocardium, this occurs due to an abrupt reduction in blood flow to the myocardium, which is then subsequently restored, but no infarct occurs [22]. However, there is dysfunction of the involved myocardium. This maybe seen on stress CMR as a dysfunctional myocardial wall fitting a coronary vascular territory in the absence of infarct. A hibernating myocardium is thought to be due to repeated stunning or chronic ischemia. Chronic hypoperfusion due to reduced coronary blood flow at rest is matched by a regional reduction in cardiac function and wall thinning [22,23]. On stress CMR, this can be seen as a region of reduced perfusion, wall thinning, and reduced myocardial contractility. A low-dose dobutamine stress CMR has been shown to improve diagnostic accuracy in determining contractile reserve, and therefore identifying a hibernating myocardium [22].

## 4. Semi-Quantitative and Fully Quantitative Stress CMR

Visual assessment is subjective and highly dependent on expertise. Semi-quantitative and fully quantitative analysis of CMR perfusion based on signal intensity (SI) curves during the first pass of gadolinium contrast are solutions to this issue (Figure 5) [24]. Analysis methods that describe characteristics of the SI profile without estimating myocardial blood flow (MBF) are typically referred to as semi-quantitative analysis. The parameter calculated has been given various terms, such as the myocardial perfusion reserve (MPR) or the myocardial perfusion reserve index (MPRI) [20]. The calculation of these parameters has varied over time. For simplicity, we will refer to this as MPR, which is calculated as the ratio of MBF at peak stress and rest [25]. The MPR assesses the vasodilatory capability in response to vasodilator stress [26]. However, the MPR calculation and technique can result in different MPR values across scanners depending on how accurately the arterial input function (AIF) has been accounted for [27]. The accurate measurement of the arterial input function is an important requirement to quantify actual myocardial blood flow. To overcome the saturation or blunting of T1 signal intensity that prevents the precise measurement of the AIF, two methods have been utilised. One is a dual-bolus method [26], and the other is a dual-sequence technique [27]. 

A fully quantitative analysis allows for the measurement of the MBF in units of millilitres of blood per minute per gram (mL/min/g) for each pixel of myocardium based on the perfusion maps by a number of different models [28,29,30]. The perfusion sequence for MBF quantitation is a dual sequence that is modified based on ECG gated saturation recovery spoiled gradient recall (SPGR) sequence (Figure 6). Within each R-R interval, this dual sequence first acquires one low-resolution AIF image at a basal slice (usually slightly more basal than the basal slice for the actual perfusion sequence), followed by the standard perfusion acquisition of two to three high-resolution images depending on the HR. Images are obtained per heartbeat over 90 heartbeats under free breathing, covering a sufficient pre- and post-contrast stage. Proton density-weighted images are acquired as the first two frames of each slice for surface coil intensity correction. Typical imaging parameters used in our centres are readout field of view (FOV) = 30–40 cm (phase FOV 0.75–0.80), voxel size = 1.7–2.0 × 2.2–2.7 mm, flip angle = 20°, repetition time (TR)/time to echo (TE) = 2.7–3.2/1.1–1.6 ms, NEX = 0.75, slice thickness = 8 mm, and parallel imaging factor = 2.

Different MBF estimating models have no significant difference regarding the diagnostic accuracy of inducible myocardial ischemia, such as the Fermi model, the uptake model, the one-compartment model, and the model-independent deconvolution [30]. Motion correction to correct for respiratory motion is preferable before analysis [20]. A reduction in stress MBF is usually caused by either obstructive CAD or coronary microvascular dysfunction (CMD), but may also result from inadequate stress [31] (Figure 2 and Figure 7). Dark rim artifacts can be distinguished from remote myocardium, and true perfusion defects by comparing quantitative stress MBF [32]. Quantitative analysis of absolute MBF can also help to further evaluate the ischemia burden in multi-vessel disease, which may be underestimated due to ‘balanced ischemia’ in visual assessment [33,34]. 

## 5. Obstructive Coronary Artery Disease

Obstructive CAD, characterised by atherosclerotic plaque accumulation in the epicardial arteries, remains a worldwide public health problem with unmet need. Both anatomical and functional assessments of obstructed epicardial arteries are important for diagnosis and management decisions. Non-invasive functional imaging for myocardial ischemia is recommended as the initial test to diagnose obstructive CAD in symptomatic patients, including stress CMR, stress echocardiography, positron emission tomography (PET), or single-photon emission computed tomography (SPECT) [35]. 

The diagnostic accuracy of stress CMR is well validated. A meta-analysis including 37 CMR studies with 2841 CAD patients showed a pooled sensitivity of 89% (95%CI 88–91%) and specificity of 76% (95%CI 75–78%) of stress CMR to diagnose CAD with ≥50% stenosis in a coronary angiogram [1]. A CE-MARC study of 752 angina patients demonstrated the superiority of a stress CME over SPECT, with a higher sensitivity (86.5% vs. 66.5%, *p* < 0.0001) and negative predictive value (90.5% vs. 79.1%, *p* < 0.0001) [36]. Furthermore, in another meta-analysis taking fractional flow reserve (FFR) as the reference standard, stress CMR also presented a much higher diagnostic odds ratio (DOR) than SPECT or stress echocardiography and a comparable diagnostic ability to PET or CCTA, but without ionizing radiation [37]. Although stress echocardiography is conveniently performed in clinical practice to evaluate myocardial ischemia, the diagnostic accuracy of epicardial CAD was inferior to stress CMR (DOR: 38 vs. 20) [38]. In addition, the absolute quantification of MBF by stress CMR correlated well with PET-derived measurement (r = 0.92, *p* < 0.001) [39].

The presence of ischemia/LGE in the stress CMR is an important marker of poor prognosis. A meta-analysis of 19 studies with 11,636 patients with suspected or known CAD showed that patients with ischemia had a higher incidence of non-fatal myocardial infarction (MI) (odds ratio [OR]: 7.7; *p* < 0.0001) and cardiovascular death (OR: 7.0; *p* < 0.0001), with a mean follow-up of 32 months [4]. Patients with LGE had a significantly increased risk of cardiovascular death than patients without LGE (OR: 2.71; *p* < 0.0001) [4]. Likewise, in patients with stable chest pain syndrome, an abnormal stress CMR (ischemia+/LGE+) suggested a >four-fold higher annual rate of acute MI and cardiovascular death, increasing the rate of coronary revascularisation as a consequence [5]. More extensive ischemic burden was independently associated with all-cause mortality (HR: 1.04; 95% CI: 1.02 to 1.07; *p* < 0.001) [40]. Quantitatively, an ischemia burden of ≥1.5 segments or ≥10% myocardium with MPR < 1.5 was proposed as a CMR indicator for revascularisation, as it most strongly predicted MACEs [41,42].

Beyond diagnosis and prognosis, the clinical utility of stress CMR has been extensively investigated, such as guiding subsequent management. Despite the recommendations for non-invasive imaging in international guidelines, invasive coronary angiography (ICA) is commonly used to confirm the diagnosis of significant CAD and determine revascularisation [35,43]. However, the benefit of percutaneous coronary intervention (PCI) in these patients is under debate. The ISCHEMIA and COURAGE trials failed to show a benefit to the PCI strategy in stable angina patients compared to optimised medical therapy (OMT) in terms of MACE [44,45]. Furthermore, a more recent trial, REVIVED-BCIS2, has similarly demonstrated that PCI had no added benefit over OMT in patients with severe ischaemic left ventricular dysfunction with obstructive CAD, which is amenable to PCI [46]. These trials therefore challenge the current strategy of PCI in patients with obstructive CAD and support an OMT strategy in managing these patients initially.

By comparing to a guideline-based approach, the CE-MARC study found that stress CMR-guided care significantly reduced the unnecessary ICA rate by 79% [47]. Furthermore, the MR-INFORM trial directly compared a stress CMR-based strategy to a FFR-based strategy in a cohort of 918 patients with typical angina [6]. Although fewer patients in the CMR group were referred for revascularisation (35.7% vs. 45.0%; *p* = 0.005), the two groups had similar rates of MACEs at 1 year (3.6% in the CMR group and 3.7% in the FFR group; *p* = 0.91). In conclusion, stress CMR as the gatekeeper manages patients with suspected CAD and is noninferior to ICA with FFR for 12-month outcomes despite lower rates of revascularisation.

Recurrent MI and cardiovascular death remain common complications in patients undergoing coronary revascularisation; therefore, the standard post-revascularisation management requires aggressive secondary preventive measures, including lifestyle modifications, antiplatelet therapies, and statin [48,49]. Risk stratification after coronary revascularisation by stress testing remains challenging. A couple of studies focused on patients with a previous coronary artery bypass graft (CABG) found that inducible ischemia and the extent of the ischemic scar were the independent risk factors of cardiovascular mortality or recurrent nonfatal MI [50,51]. Seraphim et al. recently reported that a 1 mL/g/min decrease in stress MBF and 1 unit of decrease in MPR increased the relative risk of MACEs by 156% and 61% in patients after CABG, respectively [51]. Currently, in clinical practice, stress CMR is not indicated for follow-up of CABG patients in the absence of symptoms. 

Increasingly, studies have suggested that stress CMR as the first non-invasive strategy to detect significant CAD prior to coronary revascularisation, is cost-effective over other approaches. A cost-effectiveness analysis of the Stress CMR Perfusion Imaging in the United States (SPINS) study compared five clinical strategies for patients with stable chest pain syndrome in the United States [52]. A decision analytic model using cardiovascular death or acute MI as the endpoint found that the incremental cost-effectiveness ratio (CER) for the CMR-based strategy compared with the no-imaging strategy was $52,000/quality-adjusted life years (QALY), whereas the incremental CER for the immediate ICA strategy was $12 million/QALY. The overall results supported stress CMR to be a cost-effective modality as the first-line investigation of chest pain, compared with other common imaging strategies. Another similar study from the CE-MARC study compared eight approaches from different combinations of exercise treadmill testing (ETT), SPECT, CMR, and ICA in Germany. Only two strategies were found to be cost-effective, both including CMR: (1) CMR follows a positive or inconclusive ETT followed by ICA if positive or inconclusive; (2) CMR is followed by ICA if positive or inconclusive [53]. The cost-effectiveness of a CMR-driven strategy as the first procedure has been approved in other countries, such as the United Kingdom, Switzerland, and Australia [54,55]. 

## 6. Coronary Microvascular Dysfunction

Clinical and scientific interest in CMD has been continuously increasing in recent years. CMD is characterised as an impaired flow reserve of the coronary microvasculature causing chest pain and myocardial ischemia with or without significant CAD [56,57,58]. PET has been the most extensively investigated method for a non-invasive assessment of coronary microvascular function, and normal myocardial perfusion imaging with LV regional wall motion but reduced hyperemic MBF and/or MPR may signify CMD [59]. Perfusion defects can also be observed in severe cases, and therefore, a comprehensive assessment of the flow quantification, clinical course, and imaging data is critical to distinguish CMD from epicardial CAD. The quantitative diagnostic threshold of CMD for PET is under investigation, as several extrinsic and intrinsic factors may affect the measurement of hyperemic MBF and MPR [60]. 

Stress CMR also provides diagnostic value in patients with CMD [25]. The absence of an inducible perfusion defect produces excellent accuracy to identify low-risk patients with known or suspected CAD [61], and visual interpretation of myocardial perfusion CMR is limited in detecting CMD [62]. The sensitivity of visual assessment to diagnose CMD has shown to be relatively low, at only 41% (95% CI: 27% to 57%) [63]. The advent of the quantitative assessment of MBF or MPR by stress CMR is a significant development when diagnosing CMD [31]. Similar to PET, CMD can be diagnosed when patients present with reduced stress MBF or MPR after getting an adequate hemodynamic response [31] (Figure 7). Several diagnostic criteria based on stress MBF or MPR in stress CMR have been established in few studies with favorable diagnostic accuracy, validating against invasive measurements (Table 1).

**Table 1 diagnostics-13-00524-t001:** Semi-quantitative and fully quantitative diagnostic thresholds of stress CMR for CMD.

Study	N	Reference Standard	Modality	Diagnostic Measurement	AUC	Sensitivity	Specificity
2015, Thomson et al. [64]	118	Invasive CRT	CMR (semi-quantitative)	MPR < 1.84	0.78 (95% CI: 0.68 to 0.88)	73% (95% CI: 64% to 82%)	74% (95% CI: 58% to 90%)
2019, Kotecha et al. [65]	23	IMR > 25	CMR (fully quantitative)	Stress MBF ≤ 2.19	0.73 (95% CI: 0.63 to 0.84)	71%	70%
				MPR ≤ 2.06	0.68 (95% CI 0.56 to 0.80)	44%	92%
2021, Rahman et al. [63]	75	Invasive CFR < 2.5	CMR (fully quantitative)	Visual assessment	0.58 (95% CI: 0.46 to 0.69)	41% (95% CI: 27% to 57%)	83% (95% CI: 65% to 94%)
				MPR < 2.19	0.79 (95% CI: 0.68 to 0.88)	70% (95% CI: 53% to 83%)	90% (95% CI: 74% to 98%)

CRT: coronary reactivity testing; CMR: cardiac magnetic resonance; IMR: index of microcirculatory resistance; MPR: myocardial perfusion reserve; MBF: myocardial blood flow; AUC: area under curve; CI: confidence interval.

Prognostically, the long-term outcome of CMD patients is not favorable, with increased risk of cardiovascular death, non-fatal MI, non-fatal stroke, and hospitalisation due to heart failure or unstable angina [66,67,68]. The Coronary Vasomotor Disorders International Study (COVADIS) Group looking into the prognosis of CMD reported that the annual incidence of the composite of MACEs per patient year was 7.7%, with hospitalisation for unstable angina occurring most often [69]. These results have also been confirmed in CMR studies in a non-invasive manner. The Women’s Ischemia Syndrome Evaluation (WISE) study first showed that global MPR can be predictive of MACEs in a women cohort with symptomatic myocardial ischemia without obstructive CAD [70]. For each 1 unit decrease in MPR, the adjusted HRs for death and MACE were 2.22 (95% CI, 1.16–4.23, *p* = 0.015) and 1.65 (95% CI, 1.14–2.38, *p* = 0.008), respectively [71]. Zhou et al. found a similar result and proposed the optimal cut-off value of MPR ≤1.47 in predicting MACEs in a CMD cohort of both men and women (HR = 3.14; 95% CI: 1.58 to 6.25; *p* = 0.001) [62]. Stress MBF is also an independent prognostic factor of MACEs with HR of 2.28 (95%CI, 1.43–3.66, *p* = 0.001) in patients without regional perfusion defects [71]. Taken together, these studies support the notion of assessing MBF or MPR routinely, not only for diagnosing but also stratifying patients with suspected CMD, and potentially targeting therapy in the future. 

## 7. Atrial Fibrillation

AF is a common concomitant condition in patients with CAD and likely worsens the prognosis of a patient with CAD [72]. However, the presence of AF often challenges the CMR image acquisition and interpretation of results of myocardial ischemia. Standard electrocardiographically gated CMR in patients with irregular heart rate may result in suboptimal images despite various CMR techniques to compensate for arrhythmia [73,74]. A single-shot first-pass perfusion sequence, which is less sensitive to arrhythmia and breath holding, allows for the feasibility of stress CMR to scan AF patients with good image quality [9,74]. Ungated stress CMR used in AF patients is under development; it shows a diagnostic accuracy of 96% in detecting CAD, but with reduced temporal and spatial resolution [75,76].

Currently, stress CMR is sometimes performed in AF patients with possible ischemic symptoms and the potential risk of CAD in clinical practice. However ,the standard screening strategy for CAD utilising stress CMR in AF patients has not been well established [77]. In terms of prognosis, in 539 patients with AF and suspected or known CAD, Pezel et al. showed that the presence of ischemia or LGE in stress CMR had the prognostic value to predict cardiovascular death and nonfatal MI over a median follow-up period of 5.1 years [9]. In another retrospective study with a similar cohort but a longer follow-up, Weiss et al. found patients with both ischemia and LGE had the highest cumulative rate of MACEs versus ischemia or LGE only [78]. However, whether patients with AF and imaging evidence of ischemia will benefit from coronary revascularisation is unclear [79]. Future prospective randomised studies incorporating stress CMR in AF patients will be required. 

## 8. Cardiomyopathy

Stress CMR is complementary in hypertrophic cardiomyopathy (HCM), unless CAD is suspected [80]. However, the diffuse nature of the disease generally affects coronary microvasculature of the whole heart and consequently causes myocardial ischemia [81]. An adenosine-induced CMR study of 115 HCM patients reported that visual perfusion defects were present in 41.7% patients, and none of the defects corresponded with coronary territories [82]. Subendocardial-inducible hypoperfusion localising to hypertrophied myocardial segments with multiple patchy patterns or a concentric pattern is a typical presentation of ischemia due to CMD in HCM [82,83]. Regarding the clinical significance, a perfusion defect was found to be associated with non-sustained ventricular tachycardia, higher LV mass index, and apical aneurysms [83,84].

Likewise, in dilated cardiomyopathy (DCM), vasodilator stress-rest perfusion is performed to determine the presence of inducible ischemia [13]. Even if DCM is considered a non-ischemic myocardial disease, the affected patients usually show diffuse myocardial hypoperfusion (Figure 8) [85]. A recent quantitative stress CMR study showed that DCM patients had significantly reduced stress MBF and global MPR with increased rest MBF versus normal controls, likely due to an increased hemodynamic load and structural alteration of the coronary microvasculature [86,87]. Furthermore, coronary vasodilatory dysfunction was correlated with worse LV systolic function [86]. The results from stress CMR revealed the myocardial damage in DCM was caused by an impaired coronary vasodilatory reserve, rather than chronic myocardial hypoperfusion. In the future, CMD detected by stress CMR may serve as the innovative target in DCM patients, and the long-term prognosis needs to be investigated. 

Myocardial remodeling with fibrosis replacement is a common pathophysiology in both HCM and DCM that mostly affects coronary microvasculature to cause myocardial ischemia [88]. Typical patchy and mid-wall distribution of LGE indicating myocardial fibrosis infiltration is helpful to distinguish nonischemic cardiomyopathies from advanced CAD [89]. In addition, advanced native and post-contrast T1 mapping can provide complementary information to support the detection of myocardial fibrosis. Both native T1 and extracellular volume (ECV) are prolonged in HCM and DCM patients, consistent with the presence of diffuse interstitial fibrosis and myocardial collagen content. The increased native and post-T1 values are correlated with hypertrophic or reduced wall thickness [90]. The quantitative measurement of myocardial fibrosis via T1 mapping techniques is associated with an excellent overall correlation with myocardial biopsy results [91]. 

## 9. Diabetes

Patients with diabetes experience an increased risk of cardiovascular morbidity and mortality due to several pathophysiological conditions, such as epicardial CAD, CMD, cardiac remodelling with heart failure, and peripheral vascular diseases [92]. Stress CMR has made continuous progress in assessing patients with diabetes to understand the natural history and prognosis, especially in detecting silent epicardial CAD. Ng et al. found a prevalence of 20.6% silent obstructive CAD in asymptomatic diabetic patients with Framingham risk  ≥  20% [93]. Diabetic patients had decreased global MBF during stress and increased global MBF at rest, due to insulin resistance and increased fatty acid oxidation, requiring a higher basal oxygen consumption [94]. Moreover, the presence of inducible myocardial ischemia was the strongest predictor for cardiovascular death and non-fatal MI (HR: 4.86, 95%CI 1.61–14.67, *p* < 0.01) [95]. Patients without inducible ischemia had significantly lower annual event rates than those with ischemia (1.4% vs. 8.2%; *p* = 0.003). Similarly, Kwong et al. reported that in 28% of diabetic patients, a prior unrecognised MI was found by CMR, which predicted a >four-fold increase for MACEs risk (HR: 4.13, 95% CI 1.74–9.79, *p* = 0.001) [96]. 

## 10. Obesity 

Excess adiposity prompts adverse changes in the myocardium and vasculature and developing CVDs, such as CAD, heart failure, sudden cardiac death, and AF [97]. Common stress tests, such as stress echocardiography, SPECT perfusion imaging, or CCTA, are usually limited in obese patients by suboptimal image quality due to poor acoustic windows, soft tissue attenuation artefacts, and increased noise [11,98]. In comparison, stress CMR is highly feasible and can produce diagnostic-quality imaging in more than 95% of patients [11,99]. In addition, stress CMR has excellent prognostic value in obese patients. Shah et al. found that inducible ischemia was independently associated with cardiac death or nonfatal MI (HR = 7.5; 95%CI 2.0–28.0; *p* = 0.002) [99]. Ge et al. confirmed similar results in a multicentre study with 2349 obese patients. At a median follow-up duration of 5.4 years, the annual rate of MACE was low, at ≤1% in obese patients with neither ischemia nor LGE [11]. However, in patients with ischemia or LGE, the risk of MACEs increased two to three times, which was associated with early (<90 days) referral to angiography and coronary revascularisation [11]. 

## 11. Systemic Lupus Erythematosus

Cardiac involvement is the leading cause of morbidity and mortality in patients with systemic lupus erythematosus (SLE), which is underestimated in clinical practice [100]. A comprehensive stress CMR protocol, including myocardial tissue characterisation and myocardial perfusion, may benefit patients with SLE. CMR may be preferred in SLE patients with suspected myocarditis or pericardial disease to confirm the diagnosis and to assess treatment response [101]. Meanwhile, stress CMR allows for the assessment of ischemic burden mainly due to CMD and explains the occurrence of persistent chest pain [102]. A previous study of 20 female SLE patients without obstructive CAD showed that the prevalence of abnormal stress CMR was as high as 44%, and MPR was significantly reduced, in both subendocardium and subepicardium by the effect of CMD [10]. At the 5-year follow up, 25% of patients with an impaired MPR at baseline had a lower MPR value [103]. Perfusion defects were associated with hypertension, renal disorder, repolarisation abnormalities, and increased LV size [104].

## 12. Heart Transplantation

Survival and life quality after heart transplant is associated with the occurrence of complications, such as acute rejection and cardiac allograft vasculopathy (CAV). Early identification is critical, because it may allow for alterations in medical therapy before progression [105]. Multiparametric CMR with T1, T2, and ECV quantification allows for the non-invasive detection of myocardial oedema and inflammation due to myocytes damage and haemorrhage in acute rejection [106]. Among these techniques, T2 mapping has been well validated in animal studies to show a strong correlation with actual myocardial water content in particular [107]. A combined approach of T2 mapping and ECV with increased values provides a promising diagnosis of acute rejection and avoids endomyocardial biopsy in 63% of patients [106]. 

CAV is characterised as a diffuse concentric intimal hyperplasia involving both epicardial, intramyocardial coronary arteries, and even veins, due to complicated immune-mediated pathophysiology, which is different from typical CAD with major epicardial vessels involved with focal, eccentric, and degenerative plaques [108]. The LV ejection fraction (EF), stroke volume (SV), and cardiac output (CO) may be normal either in early or in late stages, and LVEF does not correlate with the degree of CAV [109]. CCTA is useful to evaluate the coronary lumen, but it is limited by the difficulty to reach the appropriate heart rate in post-transplant patients and due to the poor visualisation of distal coronary arteries [110,111]. CCTA’s usefulness in CAV requires further research. The value of coronary artery calcium (CAC) quantification remains controversial. The absence of CAC is not reliable enough to exclude CAV, and 36.4% severe patients have been reported to have a calcium score of zero [112,113]. Due to the diffuse disease of CAV, myocardial perfusion imaging has been extensively performed in CAV. PET-derived flow parameters correlate significantly with invasive coronary flow indices. Chih’s group has shown that a combination of MPR < 2.9, stress MBF < 2.3, and coronary vascular resistance (CVR) > 55 proved a high diagnostic accuracy for CAV, validated with ICA and multivessel intravascular ultrasound (IVUS) [114,115]. Flow quantification using stress CMR therefore holds the potential for CAV diagnosis by detecting a homogenous reduction in flow [116]. Previous studies have shown that MPR assessed by stress CMR significantly outperformed ICA to detecting moderate CAV, and MPR ≤ 1.68 had a 100% sensitivity and 100% NPV in detecting CAV [117,118].

Stress CMR is also a safe technique to assess post-transplant complications, and it provides risk stratification information. Adenosine may cause an exaggerated sinus node and atrioventricular node suppression in patients after heart transplantation, so a lower dose is considered for safety concerns [119,120]. Meanwhile, regadenoson, as a newer selective A2A adenosine receptor agonist, has been proven to be safe and well tolerated in heart transplant recipients. A feasibility study reported that there were no events that required an early termination of the test, such as atrioventricular block, symptomatic arterial hypotension, or poor tolerance to the symptoms [121]. Although the hemodynamic response was attenuated in transplanted patients, the performance of the test was not affected [121]. An abnormal regadenoson stress CMR was associated with a significantly higher incidence of the composite endpoint of MI, PCI, cardiac hospitalizsation, retransplantation, and death (3-year cumulative incidence estimates of 32.1% vs. 12.7%, *p* = 0.034) [122]. 

## 13. Children

Myocardial ischemia occurs in children due to both congenital and acquired heart diseases. In children, stress CMR has been used for evaluation in a broad array of conditions, including congenital heart disease (CHD), Kawasaki disease, post-surgical assessment of coronary artery anomaly, and cardiac transplant surveillance [123,124,125,126,127]. Previous pediatric studies have shown that up to 43.5% of pediatric patients with congenital and acquired heart diseases referred for stress CMR have myocardial perfusion defects [128,129,130]. Impaired myocardial perfusion on stress CMR has been shown to correlate well with invasive FFR and coronary artery stenosis in children with coronary anomalies and Kawasaki disease [129,130].

Performing stress CMR in children presents different challenges, such as their higher heart rates, discomfort from intravenous line placement, sedation/general anesthesia, and poor cooperation with breath-holding instructions [126]. The choice of pharmacological stress agents in children is the same as in adults, but the dosage and infusion rates may vary depending on circumstances (see Table 2). Stress CMR with an adenosine infusion can be performed using the same infusion rates as in adults, and the dose is dependent on weight (i.e., 140 μg/kg/min) [131]. For patients under general anesthesia, some units have used lower adenosine infusion rates and titrated upwards (i.e., start with 110 μg/kg/min, increase stepwise to 125 μg/kg/min and then 140 μg/kg/min based on hemodynamic changes) [125]. Regadenoson has fewer side effects and a simpler single bolus injection through a single peripheral intravenous cannula. Thus, this is widely used in pediatrics [126,128,132]. For pediatric patients weighing less than 40 kg, a dose of 6–10 μg/kg is recommended for safety concerns [126,128,131,133]; those ≥ 40 kg can receive the usual adult dose of 400 mcg of regadenoson [128,132]. Lastly, dobutamine can also be utilised with typical doses of 5–10 μg/kg/min with an increase every 3–5 min to a maximum dose of 40 μg/kg/min [131]. Additional atropine bolus up to 0.01 mg/kg can be given to reach 80% of maximal age predicted heart rate [134,135,136]. Dipyridamole is far less used in children, and only a few early studies have reported its feasibility [131,137]. During the entire period of pharmacological stress, close monitoring for adverse events such as heart block, bronchospasm, and hypotension should be performed. The suggested protocols of different pharmacologic agents are summarised in Table 2.

**Table 2 diagnostics-13-00524-t002:** Suggested stress CMR protocols with different pharmacological agents in children and adults.

Pharmacologic Agents	Mechanism of Action	Administration	Dose/Infusion Rate	Known Side Effects	Recommendation
Adenosine [13,125,131,138,139]	Endogenous vasodilator affects A1/A2A receptor	IV infusion; 2 PIV	Children: 140 μg/kg/min Adult: 140 μg/kg/min and increase up to 210 μg/kg/min without adequate stress response	Mild tachycardia, chest discomfort, flushing, and nausea	Recommend in both children and adults
Regadenoson [13,126,128,131,132,133]	Selective cardiac A2A adenosine receptor agonist	IV infusion; 1 PIV	Children: <40 kg: 6–10 mcg/kg ≥40 kg: 400 mcg Adult: 400 mcg	Limb tingling, nausea/gastrointestinal discomfort, anxiety, chest pain, mild flushing, and mild headache	Recommend in both children and adults
Dobutamine [13,134,135,136]	Synthetic catecholamine stimulates beta-1 receptors in the heart	IV infusion; 2 PIV	Children: start at 5–10 mg/kg/min with an increase every 3–5 min to a maximum dose of 40 μg/kg/min Adult: start at 10 mg/kg/min with an increase every 3 min to a maximum dose of 40 μg/kg/min	Arrhythmia, chest pain, palpitations, skin rash, anxiety, dizziness, dyspnea, nausea, and emesis	Recommend in both children and adults
Dipyridamole [131,137]	Inhibition of the degradation of cyclic adenosine monophosphate	IV infusion; 2 PIV	Children: 0.142 μg/kg/min Adult: 0.142 μg/kg/min	chest pain, headache, and dizziness	Recommend in adults
ATP [13]	ATP decomposes into adenosine to promote vasodilation	IV infusion; 2 PIV	Children: not reported Adult: 140 μg/kg/min and increase up to 210 μg/kg/min without adequate stress response	flushing, chest pain, palpitations, and breathlessness	Recommend in adults

IV: intravenous; PIV: peripheral intravenous catheter; CMR: cardiac magnetic resonance; ATP: adenosine triphosphate.

## 14. Future

Stress CMR is limited in clinical practice due to the long scan time necessary; thus, a faster and more cost-effective approach is required in the future. Rijlaarsdam-Hermsen et al. proposed a new strategy to diagnose CAD and make a decision for revascularisation, which applied the coronary artery calcium (CAC) score as the gatekeeper for subsequent stress-only CMR. First, the result demonstrated that stress-only CMR was promising for detecting obstructive CAD, as the sensitivity was 90.9% (95% CI: 88.7 to 93.1), and the specificity was 98.7% (95% CI: 97.9 to 99.6). Second, the percentage of patients with obstructive CAD was related to the CAC score, where 52% exhibited significant stenosis with high CAC scores (≥400), but it was only 20% in such patients with low CAC scores (between 0.1 and 100). Lastly, the approach suggested that asymptomatic patients with low CAC scores could be deferred from further stress testing; however, patients with a CAC score ≥400 should be referred to undergo stress-only CMR, regardless of the type of chest pain. Although the cost effectiveness of this screening strategy has not been determined, the study showed the feasibility of skipping rest perfusion without influencing the diagnostic capability [140]. 

Beyond optimising the acquisition protocol, another avenue to shorten scanning time is to acquire three-dimensional (3D) cine imaging and LGE, but this requires an acceptable balance between spatial resolution, temporal resolution, signal- and contrast-to-noise, artifacts, acquisition, and reconstruction times [141]. More recently, Gómez-Talavera reported a protocol comprising isotropic 3D cine (enhanced sensitivity encoding (SENSE) by Static Outer volume Subtraction (ESSOS)) and isotropic 3D LGE sequences within two breaths held [141]. The mean acquisition time for single-breath 3D cine imaging and LEG was 24 s and 22 s, respectively, which is much shorter than 2D acquisition requiring 280 s. Cardiac function parameters and LGE assessment between 3D and 2D CMR had an excellent agreement and insignificant bias. This full 3D study in a door-to-door time of <15 min removes the time-consuming planning of imaging planes and can compete with echocardiography as the first-line testing method in many clinical settings. Incorporating this advanced technique with dynamic contrast-enhanced perfusion imaging may fundamentally change the clinical utility of stress CMR. 

Advancing stress imaging acquisition is another breakthrough. Stress CMR T1 mapping is an innovative technique to measure the increased myocardial blood volume in the myocardium during stress through the partial volume of blood T1 [142,143]. Under normal conditions, the myocardium demonstrates a normal T1 value at rest, which increases due to coronary vasodilation induced by stress agents [144]. However, a pathophysiology change in coronary circulation will result in the abnormal relaxation of T1 time. Alexander et al. validated this technique in CAD patients and found that infarcted myocardium and ischemic myocardium had no significant T1 reactivity compared with remote myocardium, and the infarcted area had the highest resting T1 of all tissue classes [144]. Stress/rest T1 mapping can differentiate between normal, infarcted, ischemic, and remote myocardium with distinctive T1 profiles without injecting gadolinium contrast. In addition, in the absence of obstructive CAD, the percentage change in the T1 value from rest to stress is likely to reflect coronary vascular reactivity. Levelt et al. showed that diabetic patients presented blunted relative stress T1 response versus normal controls due to CMD. Stress T1 mapping enables the early detection of such subclinical circulatory abnormalities, providing an opportunity for early therapeutic intervention in the future [145].

## 15. Conclusions

Qualitative and quantitative stress CMR enables one-stop assessment of myocardial ischemia, myocardial viability, global and regional cardiac function, MPR, and MBF at stress and rest. It has been widely used to evaluate patients with suspected or known significant CAD and CMD. It has shown superiority to other established non-invasive stress testing methods regarding cost-effectiveness as the first-line evaluation for chest pain. It can also predict MACEs in various CVDs, guide the decision for revascularisation, and provide clinical risk stratification. Given its various advantages, stress CMR has been increasingly used for evaluations in other clinical settings, such as cardiomyopathy, connective tissue diseases, metabolic syndrome, and post heart transplantation.

## Figures and Tables

**Figure 1 diagnostics-13-00524-f001:**
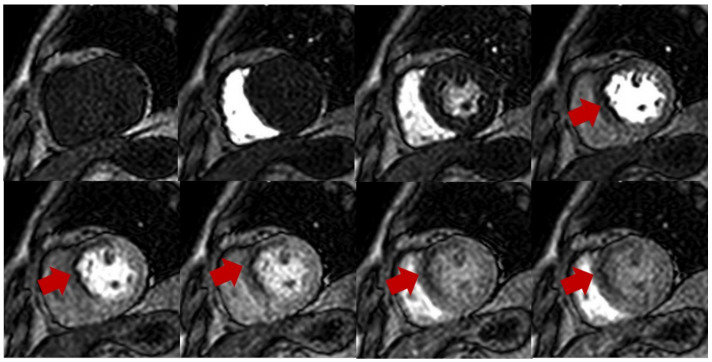
Stress perfusion showing myocardial ischemia using adenosine. Upper and lower panel shows a single basal ventricular short axis slice acquired during stress perfusion imaging. The gadolinium contrast first enters the right ventricle, then the left ventricle, and finally perfuses the myocardium. Red arrow shows myocardial ischemia in the basal anteroseptal wall with decreased T1 signal compared with normal segments.

**Figure 2 diagnostics-13-00524-f002:**
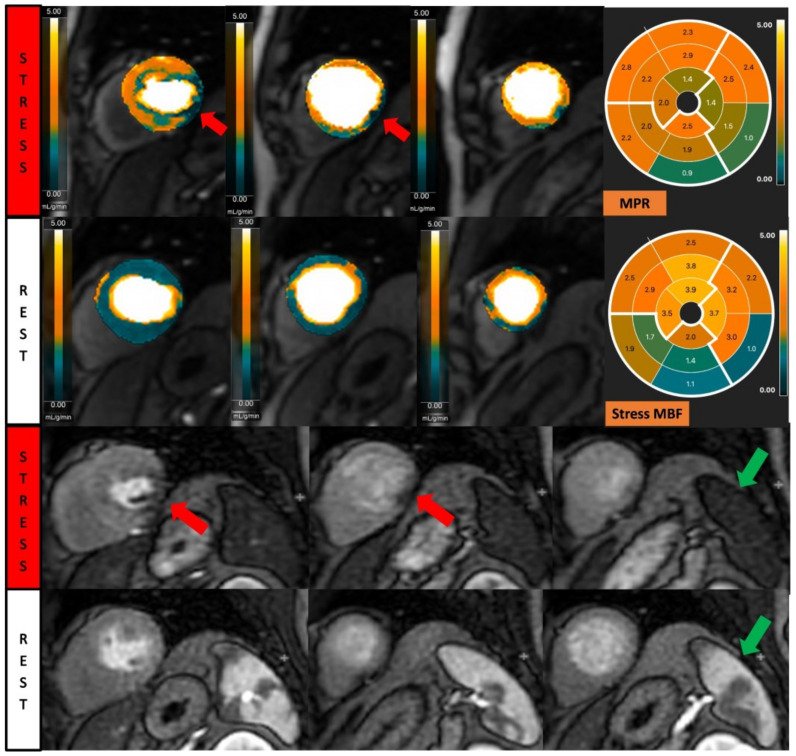
Fully quantitative analysis of myocardial blood flow (MBF) of significant CAD. Male, 55 years old, presenting with chest pain during hiking. CCTA showed chronic total occlusion in left circumflex artery (LCx). Quantitative stress perfusion images (top row of images) and standard stress perfusion images (2nd bottom row) showed hypoperfusion in the lateral wall from base to mid-ventricular slice (red arrow), corresponding with reduced stress MBF in LCx-supplied area. Green arrows highlight the splenic switch-off sign, with the spleen showing reduced perfusion during stress and increased perfusion during rest.

**Figure 3 diagnostics-13-00524-f003:**
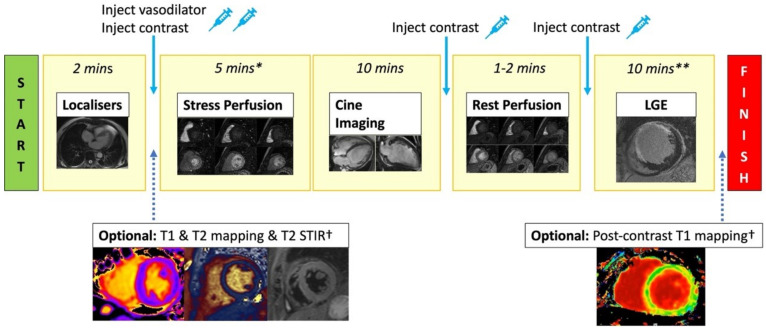
Standard stress CMR protocol. Stress CMR usually starts with acquiring localisers. T2 STIR, T1 map, and T2 map are optionally performed. Injecting vasodilator to achieve adequate stress is required before performing stress first-pass perfusion imaging (3 short axis slices). Gadolinium-based contrast is injected when adequate stress has been achieved. Next, cine imaging in the short axis and long axis views are acquired to assess cardiac anatomy and function. Rest perfusion imaging with contrast injection is then repeated. A further bolus of contrast is given, and after a 5 min delay, LGE images across the whole heart are acquired to assess for viability. * = includes time for vasodilator infusion/injection, although this time varies depending on the agent used. ** = includes delay time after contrast injection. † = optional sequences if included will increase the scanning time.

**Figure 4 diagnostics-13-00524-f004:**
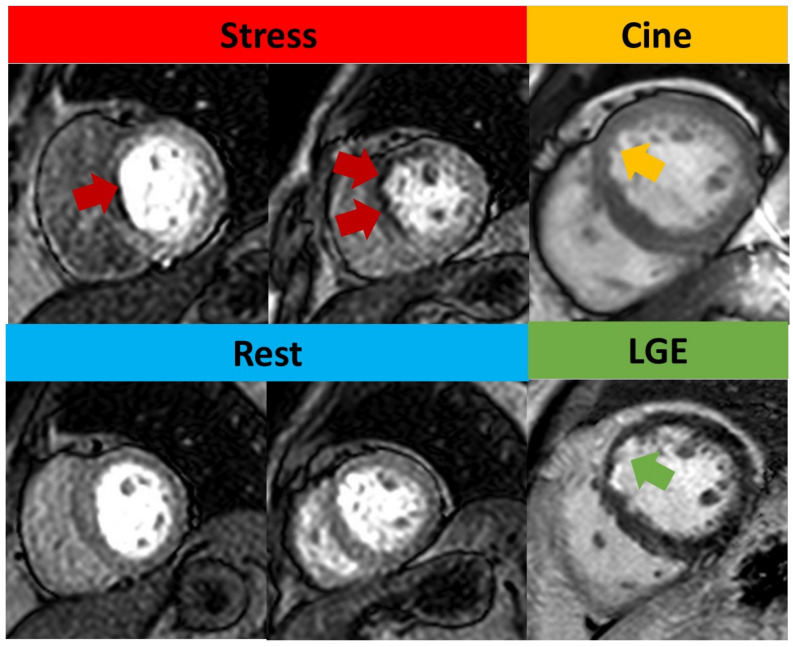
Qualitative assessment of stress CMR. Subendocardial perfusion defects presented on the basal septum and mid-anteroseptal wall (red arrow) but with normal rest perfusion. Cine image shows thinner myocardium (yellow arrow) compared to remote myocardium. LGE shows subendocardial enhancement (green arrow), consistent with left anterior descending artery (LAD) infarction. As this infarct involves <50% of the wall thickness, this was regarded as viable.

**Figure 5 diagnostics-13-00524-f005:**
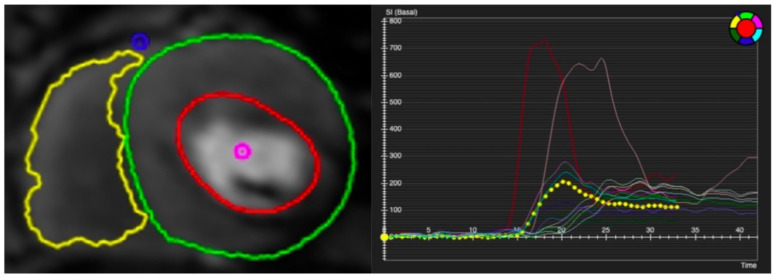
Contoured perfusion image (**left**) and signal intensity graph (**right**). Left panel demonstrates the contouring of right ventricle (yellow circle) and epicardium (green circle) and subendocardium (red circle) of left ventricle. Right panel shows the signal profile of each segment and blood pool.

**Figure 6 diagnostics-13-00524-f006:**
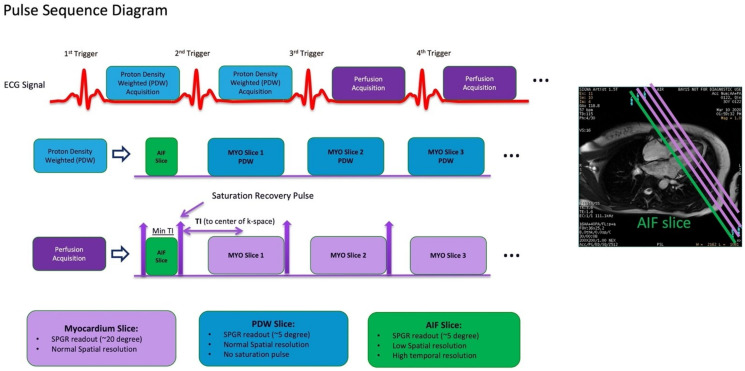
Dual sequence for quantitative myocardial perfusion. Proton density-weighted images are acquired as the first two frames of each slice for surface coil intensity correction. After each R-wave, the arterial input function (AIF) is acquired first, followed by the standard short axis slices. MYO = myocardium. SPGR = spoiled gradient recall.

**Figure 7 diagnostics-13-00524-f007:**
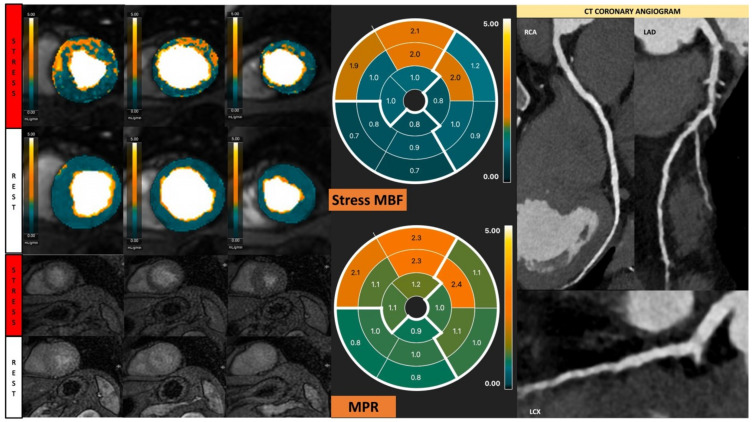
Fully quantitative analysis of stress CMR examination showing coronary microvascular dysfunction. Male, 66 years old, asymptomatic with past history of diabetes, hypertension, and hyperlipidemia. CCTA showed non-obstructive CAD, and stress CMR showed a splenic switch-off sign and no visual perfusion defect during stress. However, fully quantitative analysis showed extensive reduced MPR, not correlated to a coronary distribution, which may be caused by CMD.

**Figure 8 diagnostics-13-00524-f008:**
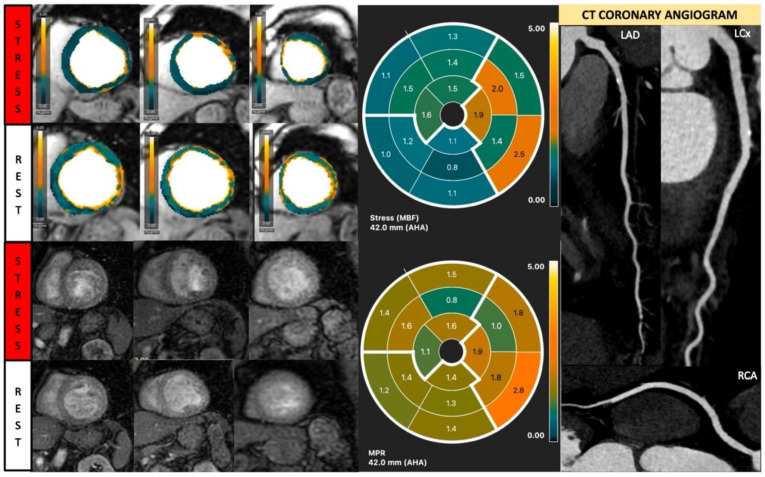
Quantitative stress CMR examination of dilated cardiomyopathy. Female patient in her 60s with dilated cardiomyopathy and phospholamban genetic mutation. Patient underwent stress CMR, having presented with symptoms and signs of heart failure. Top left panel of six images shows quantitative perfusion images with reduced global myocardial blood flow (MBF) during stress. Lower left panel of stress perfusion images shows no stress-induced perfusion defect. Stress MBF and myocardial perfusion reserve bulls-eye plots objectively demonstrate this low perfusion. Coronary computed tomography multiplanar reformat images of the three main coronary arteries show no significant coronary artery disease.

## Data Availability

Data are available on request to the author.

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
