# Peer review of "Qualitative and Quantitative Stress Perfusion Cardiac Magnetic Resonance in Clinical Practice: A Comprehensive Review"

_diagnostics, 2023, doi:10.3390/diagnostics13030524_

Round 1

Reviewer 1 Report

1.The clinical implications and usefulness  of stress CMR in obstructive CAD should be discussed as they relate to the findings of ISCHEMIA and REVIVED-BCIS2, two pivotal trials which I could not find in the references.

2.Further to my previous comment why should one prefer stress CMR and not stress ECHO for the evaluation of obstructive CAD in view of the findings of these two trials ?

3. Coronary microvascular dysfunction: several non-invasive (e.g. transthoracic Doppler-echocardiography assessing CFR, CMR, PET) and invasive methods (e.g. evaluation of CFR and microvascular resistance (MVR) using adenosine, microvascular coronary spasm with acetylcholine) have been used for the assessment of coronary microvascular function. However, the combined invasive assessment of coronary vasoconstrictor as well as vasodilator abnormalities (interventional diagnostic procedure, IDP) represents the most comprehensive coronary vasomotor assessment. Any comments?

4. Cardiomyopathies: Would the authors prefer stress CMR  to detect perfusion defects or CMR techniques to detect fibrosis?

Author Response

Response to the reviewer 1 comments:

Point 1: The clinical implications and usefulness of stress CMR in obstructive CAD should be discussed as they relate to the findings of ISCHEMIA and REVIVED-BCIS2, two pivotal trials which I could not find in the references.

Response 1: Thanks for providing us with these two important trials. We have added the related findings in Line 266-273. As discussed, the results from the trials showed that direct invasive strategy added no benefit over optimal medical therapy in patients with stable coronary artery disease (CAD). Direct invasive coronary angiography screening may not be appropriate for ischemic patients. Therefore, non-invasive stress CMR is needed as the gatekeeper and possibly to guide subsequent revascularization or optimised medical therapy.  

Point 2: Further to my previous comment why should one prefer stress CMR and not stress ECHO for the evaluation of obstructive CAD in view of the findings of these two trials ?

Response 2: Thanks for your further question. These two trials did not compare stress CMR and stress ECHO. A meta-analysis cited in this paper including thirty-seven studies and 4721 vessels has compared the diagnostic value of different non-invasive stress testing and showed that echocardiography is less accurate in ruling out hemodynamically significant CAD  than stress CMR[1]. Another meta-analysis comparing the diagnostic accuracy of stress CMR and dobutamine stress echocardiography also showed that CMR has a superior diagnostic test accuracy over echocardiography (sensitivity [0.88 (95% CI: 0.85-0.90) vs. 0.72 (95% CI: 0.61-0.81)]; diagnostic odds ratio [38 (95% CI: 29-49) vs. 20 (95% CI: 9-46)])[2]. Please see the related details in Line 240-246. Besides, stress CMR with high spatial resolution can produce better image quality than echocardiography. Therefore, stress CMR is preferred than stress echo to evaluated obstructive CAD.

Point 3: Coronary microvascular dysfunction: several non-invasive (e.g. transthoracic Doppler-echocardiography assessing CFR, CMR, PET) and invasive methods (e.g. evaluation of CFR and microvascular resistance (MVR) using adenosine, microvascular coronary spasm with acetylcholine) have been used for the assessment of coronary microvascular function. However, the combined invasive assessment of coronary vasoconstrictor as well as vasodilator abnormalities (interventional diagnostic procedure, IDP) represents the most comprehensive coronary vasomotor assessment. Any comments?

Response 3: Thanks for raising a good question. Coronary microvascular dysfunction (CMD) has complicated mechanisms, which can be caused by both impaired vasodilatory function and enhance microvascular vasoconstriction/spasm. Therefore, a combined assessment of different methods is needed to address the underlying mechanisms of CMD, such as using adenosine to assess vasodilatory microvascular function and performing acetylcholine testing to detect coronary spasm. Stress CMR can assess vasodilatory microvascular function but not vasospasm. So, stress CMR will need to be combined with other tests, such as invasive acetylcholine challenge. The interventional diagnostic procedure work-up combined with non-invasive assessment (PET/CMR/Echo) to differentiate the CMD or coronary vasospasm may facilitate more tailored treatments clinically.  

Point 4: Cardiomyopathies: Would the authors prefer stress CMR to detect perfusion defects or CMR techniques to detect fibrosis?

Response 4: Stress CMR for detecting perfusion defects and CMR for detecting fibrosis can be done simultaneously. So preferring one or the other will depend on the clinical context but can also be utilsied as a one-stop shop assessment. Perfusion defects in cardiomyopathies may be the consequence of epicardial CAD and CMD. The visual perfusion defects corresponding with coronary territories and subendocardial LGE are important CMR features to work out possible epicardial CAD. However, myocardial fibrosis affecting coronary microvasculature is more common in cardiomyopathies to cause myocardial ischemia. Multiple patchy pattern or concentric pattern of perfusion defects in the subendocardium and non-ischemic LGE pattern in the mid ventricular wall are typical presentation. In addition, T1 mapping techniques allow the quantitative CMR assessment of diffuse myocardial fibrosis with increased native T1 and ECV values. Taken together, stress perfusion imaging, LGE and native and post T1 mapping can comprehensively detect fibrosis in the context of cardiomyopathies non-invasively, and the results of T1 techniques are well validated histopathologically[3]. We have added the related contents in the revised paper to better understand the value of stress CMR and mapping techniques. Please see the details in Line 403-413.

  1. Perera, D., et al., Percutaneous Revascularization for Ischemic Left Ventricular Dysfunction. New England Journal of Medicine, 2022. 387(15): p. 1351-1360.
  2. Haberkorn, S.M., et al., Vasodilator Myocardial Perfusion Cardiac Magnetic Resonance Imaging Is Superior to Dobutamine Stress Echocardiography in the Detection of Relevant Coronary Artery Stenosis: A Systematic Review and Meta-Analysis on Their Diagnostic Accuracy. Front Cardiovasc Med, 2021. 8: p. 630846.
  3. Diao, K.Y., et al., Histologic validation of myocardial fibrosis measured by T1 mapping: a systematic review and meta-analysis. J Cardiovasc Magn Reson, 2016. 18(1): p. 92.

Reviewer 2 Report

This is a review article regarding detection of ischemia using cMRI. This reviewer considers that this paper was well written, but has some comments as described below. 

Major comments:

1.     Figure 3. This reviewer feels that this protocol is too short. Is it a generally used protocol? This figure shows that LGE needs 10 minutes, but it might need more time. Does it depend on the MRI model? 

2.     The authors should focus more on T1 and T2 mapping, as well as the differential imaging between cardiomyopathy and coronary artery disease, especially coronary microvascular diseases. 

Author Response

Response to Reviewer 2 comments:

Point 1: Figure 3. This reviewer feels that this protocol is too short. Is it a generally used protocol? This figure shows that LGE needs 10 minutes, but it might need more time. Does it depend on the MRI model?

Response 1: Thanks for your questions. In our centre, we use newer MRI scanners with single shot late gadolinium enhancement sequences which allow rapid acquisition of images. Furthermore, as gadolinium-based contrast agents have already been given for the stress and rest perfusion images, the wait time after contrast injection can be reduced to no more than 5 minutes. (Society of Cardiovascular Magnetic Resonance 2020 Guidelines)

Point 2: The authors should focus more on T1 and T2 mapping, as well as the differential imaging between cardiomyopathy and coronary artery disease, especially coronary microvascular diseases.

Response 2: Thanks for your suggestion. As this review paper focuses on the stress perfusion CMR, we talk less on T1 and T2 mapping. However, we have added some contents in the related clinical implications such as cardiomyopathies and heart transplantation, where T1 and T2 mapping techniques are well used. Please see Line 307-413/474-478/575-590 for details. In terms of the differential imaging between cardiomyopathy and CAD, it can be distinguished from both perfusion images and LGE images. We have covered this in several places. Firstly, we have presented how to diagnose myocardial ischemia due to epicardial CAD in the part of Interpretation of Stress Perfusion CMR (Line 128-144). We then emphasised the image characterization of cardiomyopathies in Line 386-388/405-407. A complementary description of T1 mapping techniques has been added in the revised manuscript, which can provide more pathological information of cardiomyopathies. Please see Line 407-413 for details. 

Reviewer 3 Report

I congratulate the authors for a clear and concise review of Qualitative and Quantitative Stress Perfusion Cardiac Magnetic Resonance in Clinical Practice: A Comprehensive Review.

Strength:

Literature Review is good

The figures and the method described for the CMR perfusion are excellent.

Few points to improve:

The title should state that the study is for adults adults

At the same time: make a small subsection or add in each section: its role in children.

In differentiating myocardial viability, please describe in depth how to differentiate myocardial stunning or hibernation and predict recovery.

Make a comparable Table of Prefusion CMR vs. PET for its value to differentiate myocardial blood flow and small vessel disease.

Add some points on the composite role of coronary calcium score, myocardial perfusion, LVEF, and sensitivity in detecting CAV in heart transplant patients and how this differs from CAD. 

Author Response

Response to Reviewer 3 comments:

Point 1: The title should state that the study is for adults.

Response 1: Thanks for your suggestion. As we have added a subsection of stress CMR in children according to your second comment, it should be better to keep the original title.

Point 2: Make a small subsection or add in each section: its role in children.

Response 2: Thanks for your suggestion to improve our work. We have added a subsection of stress CMR application in children and created a comparable table of different pharmacological agents in both children and adults. Please see Line 512-545 and Table 2 for details.

Point 3: In differentiating myocardial viability, please describe in depth how to differentiate myocardial stunning or hibernation and predict recovery.

Response 3: Thanks for your suggestions. Myocardial stunning and hibernation are two types of myocardial ischemia with viable myocardium, which may present normal signal intensity in LGE. However, we can make a differential diagnosis from myocardial perfusion, resting coronary artery flow and clinical course. We have added some points accordingly. For recovery prediction, LGE provides useful information to guide revascularization. Please see the details in Line 144-150/151-162.

Point 4: Make a comparable Table of Prefusion CMR vs. PET for its value to differentiate myocardial blood flow and small vessel disease.

Response 4: Thanks for your suggestion. We have extensively searched the literatures regarding this comment; however, we did not find studies to compare stress CMR with PET directly to diagnose small vessel disease based on MBF or MPR. We have found several CMR studies establishing diagnostic thresholds of CMD and presenting its diagnostic capability, and we have added these studies in the manuscript (Table 1). Whereas for PET, there is a lack of studies for determining the diagnostic accuracy for CMD quantitatively, and the diagnostic cut-off value of MBF or MPR is under investigated. We have pointed out this interesting finding in Line 315-323. Therefore, we are sorry for not making such a comparable table of Stress CMR vs. PET, which warrants further investigations in the future.

Point 5: Add some points on the composite role of coronary calcium score, myocardial perfusion, LVEF, and sensitivity in detecting CAV in heart transplant patients and how this differs from CAD.

Response 5: Thanks for your suggestions. We have further clarified in Line 479-498 and covered coronary calcium score, myocardial perfusion LVEF and sensitivity in detecting CAV in heart transplant patients and how this differs from CAD. We have also mentioned that quantitative stress perfusion imaging may hold the promise to diagnose CA but needs further validation in the future.

Reviewer 4 Report

Dear authors

Thanks for outstanding review of CMR. Although very logical description, another non-invasive stress test, dobutamine stress echocardiogram or nuclear imaging (Heart SPECT), should be compared. Please add comparison to CMR and different stress test. 

Best regards

Author Response

Response to Reviewer 4 comments:

Point 1: Thanks for outstanding review of CMR. Although very logical description, another non-invasive stress test, dobutamine stress echocardiogram or nuclear imaging (Heart SPECT), should be compared. Please add comparison to CMR and different stress test.

Response 1: Thanks for your suggestions. We have compared the diagnostic value of stress CMR with other common stress tests, such as SPECT/PET/echo/CCTA. Please see the related contents in Line 235-247. Overall, stress CMR is comparable to PET or CCTA to rule out epicardial CAD but superior to stress echo and SPECT with higher diagnostic odds ratio and sensitivity. Besides, we have added some contents for different non-invasive tests in other diseases, such as coronary microvascular disease (Line 315-334) and heart transplantation (Line 484-498).

Round 2

Reviewer 2 Report

This reviewer has no further comment.